# Age-dependent ACE2/TMPRSS2 expression and SARS-CoV-2 household transmission in Gran Canaria

Jesús Poch-Páez[1†‡], Yeray Nóvoa-Medina [1,2‡*], Abián Montesdeoca-Melián [3‡],
Araceli Hernández-Betancor[4], Francisco J. Rodríguez-Esparragón[5],
Svetlana Pavlovic-Nesic[1,2], Melisa Hernández-Febles[6], Jesús M. González-Martín [5],
Laura Cappiello[5], Valewska Wallis-Gómez[1], Joaquin Quiralte-Castillo[1],
Alejandro Maján-Rodríguez[1], Martín Castillo De Vera[3], Maria T. Angulo-Moreno[1],
Augusto González-Pérez[1], Asunción Rodríguez[1], Zelidety Espinel-Padrón[1],
Elisa M. Canino-Calderín[1], Irina Manzano-Gracia[1], Elena Colino-Gil [1],
Ana I. Reyes Dominguez[1], Irina Moreno-Afonso[1], Raquel McLaughlin-García[1],
Maria L. Naranjo-Báez[3], Ana Bordes-Benitez[6], Isabel De Miguel-Martínez[4],
Carlos Rodríguez-Gallego[7], Luis Peña-Quintana[1,2,8]

**1** Department of Pediatrics, Complejo Hospitalario Universitario Insular Materno Infantil de Canarias (CHUIMI), Las Palmas de GC, Canary Islands, Spain, **2** Canarian Association for Pediatric Research (ACIP Canarias; www.acipcanarias.org), Las Palmas de GC, Canary Islands. Spain, **3** Pediatric COVID19 Team, Canarian Health Service, Las Palmas de GC, Canary Islands, Spain, **4** Microbiology Department, Complejo Hospitalario Universitario Insular Materno Infantil de Canarias (CHUIMI), Las Palmas de GC, Canary Islands, Spain, **5** Research Unit, Universtiy Hospital Dr. Negrín, Las Palmas de GC, Canary Islands, Spain, **6** Microbiology Department, Universtiy Hospital Dr. Negrín, Las Palmas de GC, Canary Islands, Spain, **7** Immunology Department, Universtiy Hospital Dr. Negrín of Gran Canaria, Las Palmas de GC, Canary Islands, Spain, **8** CIBEROBN, University of Las Palmas de Gran Canaria, Las Palmas de GC, Canary Islands, Spain

‡ The three first authors (JP, YN, AM) contributed equally to this work, and deserve the same credit for authorship.
† Deceased.
* yeraynm@hotmail.com

## Abstract

### Background

This study aimed to assess whether the expression of ACE2 and TMPRSS2 is associated with susceptibility to and severity of COVID-19 across age groups. We also evaluated the role of children in household transmission of SARS-CoV-2.

### Methods

We conducted a cross-sectional observational study including 258 households in Gran Canaria between March 10 and June 2, 2020. A total of 650 individuals (including 89 children under 18 years of age) were evaluated using a combined serological testing strategy to confirm past SARS-CoV-2 infection. Gene expression of ACE2 and TMPRSS2 was quantified from saliva samples. Demographic, clinical, and household exposure data were collected for analysis.

**Data availability statement:** The datasets generated and/or analyzed during the study are not publicly available due to participant confidentiality, but are available from the corresponding author upon reasonable request. Also, the ethics committee can be contacted at this email address: ceimprovlpa.scs@gobiernodecanarias.org.

**Funding:** This research was funded by Fundación Disa, project number OA20/024. Santa Cruz de Tenerife, Canary Islands, Spain (2020). The funders had no role in the study design, data collection and analysis, decision to publish or manuscript preparation.

**Competing interests:** The authors declare no competing interests.

**Abbreviations:** SARS-CoV-2, Severe acute respiratory syndrome coronavirus type 2; ACE2,Angiotensin-converting enzyme type 2; TMPRSS2, Type II transmembrane serine protease; RT-PCR, Reverse-transcription polymerase chain reaction; BMI, Body mass index.

## Results

The combined serological approach increased diagnostic sensitivity by 10%. Antibody levels decreased with age in children but increased with age and disease severity in adults. ACE2 expression was slightly elevated in younger children; however, after correction for multiple comparisons, there was no statistically significant association between ACE2 expression and age, antibody titers, or symptom severity.. TMPRSS2 expression did not correlate with any studied variable. Children were less frequently infected (OR = 0.56), and when infected, they experienced milder symptoms and reduced disease severity. Risk factors for transmission included older age and sharing a bedroom with the index case. In adults, risk increased with age; in children, younger age was associated with higher transmission risk.

## Conclusions

Our findings do not support a strong relationship between ACE2 or TMPRSS2 expression levels and susceptibility to or severity of COVID-19. Children appear to be less susceptible to SARS-CoV-2 infection and tend to experience a milder disease course.

---

## Introduction

Understanding the differences in disease susceptibility to the severe acute respiratory syndrome coronavirus type 2 (SARS-CoV-2) and disease severity across age groups remains a critical question in COVID-19 research. Collaborative studies have highlighted the role the immune system plays in the susceptibility to severe COVID19 disease, particularly defects in type I interferon (IFN) responses [1,2]. However, the reported defects only explain the increased susceptibility of a small percentage of the cases.

To initiate its infective cycle, SARS-CoV-2 binds to the angiotensin-converting enzyme type 2 (ACE2) and utilizes the type II transmembrane serine protease (TMPRSS2), both of which are expressedin the nasopharyngeal epithelium and other tissues [3] (Image 1). Their expression varies by anatomical site, with the highest levels observed in the nasal epithelium, followed by the oral cavity and lower expression in the lower airway [4,5]. Some studies suggest a lower ACE2 expression in children compared to adults, potentially contributing to reduced susceptibility [4,6,7].

Early in the pandemic, assumptions based on other respiratory viruses (e.g., influenza) led to concerns that children might play a significant role in SARS-CoV-2 transmission. However, published data point toward similar [8] or even lower infection compared to adults [9,10]. Transmission in pediatric cases often occurs through close household contacts and children typically experience milder symptoms [11,12]. While age has been consistently identified as a key factor influencing transmission risk [9], the impact of other factors, such as viral load and receptor expression, are still unclear [13,14]. Additionally, later variants such as Delta and Omicron have also

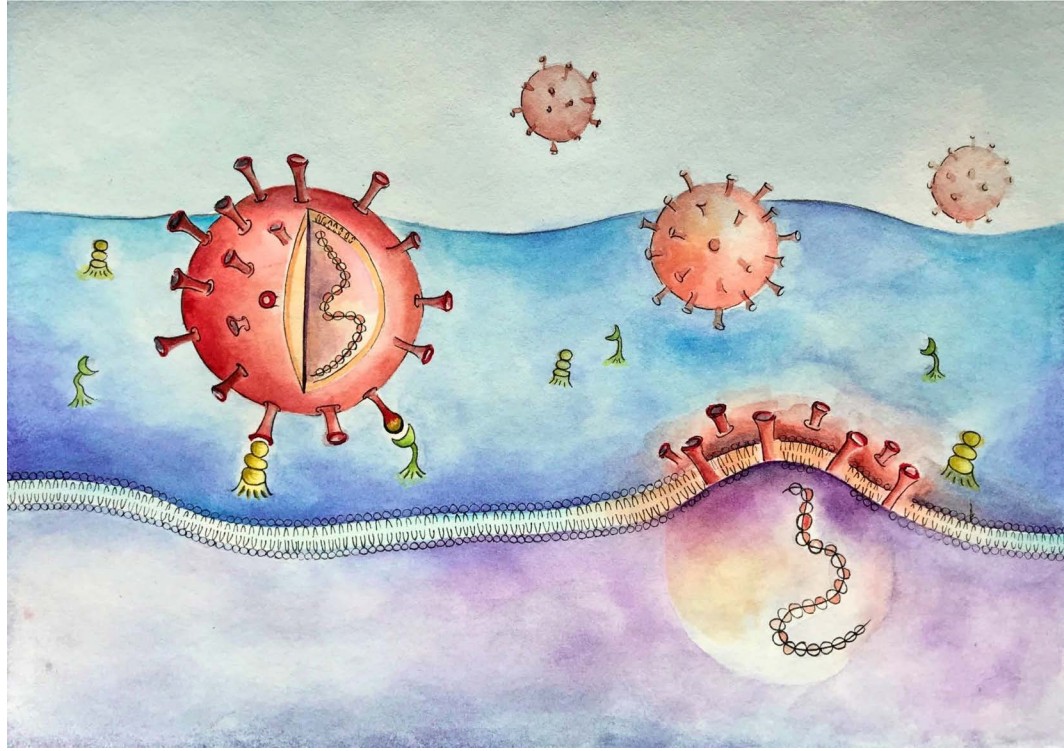

**Image 1. The interaction of ACE2 and TMPRSS2 with SARS-CoV-2.** The spike protein (S) of the coronavirus binds to ACE2 receptors and, after cleavage by TMPRSS2, enables the fusion of the cellular and viral membranes, allowing the entrance of the virus' RNA in the host cells. Original drawing. Not previously published. The Author: Anica Tengelmann Meyer, has given authorization for publication.

been shown to influence transmission dynamics, infectivity, and disease severity in children, as demonstrated by Clark et al [15] and other authors [16].

The study of SARS-CoV-2-specific antibody responses is essential for understanding both transmission and population-level immunity [17]. Most immunocompetent individuals develop detectable antiviral antibodies within two to three weeks after the onset of symptoms [18], with children presenting higher specific antibody levels than adults, and persisting longer [19]. Also, the antibody response is influenced by the variant of SARS-CoV-2, with children infected with the Omicron variant presenting lower antibody levels and function compared to those infected with the original Wuhan or Delta variants [20].

In this study, we aimed to further investigate the mechanisms underlying viral transmission by evaluating the expression of ACE2 and TMPRSS2 and their relationship with SARS-CoV-2 susceptibility, particularly in pediatric populations. Our study focused on household transmission in the island population of Gran Canaria, Spain. Gran Canaria is the third largest of the Canary Islands, covering approximately 1,560 km², with a population of about 875,000 as of 2025. Children under 18 years make up an estimated 15–18% of the population (roughly 130,000–160,000), with the remainder being adults. This setting provides insights into viral dynamics in a controlled, semi-closed environment.

## Methods

### Study population

We included 549 individuals with confirmed SARS-CoV-2 infection by reverse-transcription polymerase chain reaction (RT-PCR) between March 10 and June 2, 2020, when the ancestral Wuhan strain was the dominant circulating variant

in Gran Canaria, Spain, along with their household members. No formal sample size calculation was performed for this study. The sample size was determined by the number of participants who agreed to participate during the recruitment period. While we aimed to enroll as many participants as possible to maximize statistical power and minimize the risk of type II error, we acknowledge that the absence of a priori sample size estimation may limit the generalizability and statistical robustness of our findings. Cases were identified through the epidemiological registry of the Canary Islands Health Service. Eligible patients had confirmed infection based on clinical symptoms and epidemiological contact (e.g., travel to high-risk areas or exposure to a known case). All patients were contacted by phone, regardless of disease severity, and were invited to participate in the study along with their household members. Participation was voluntary, and written informed consent was obtained from all participants; for minors, consent was provided by a legal guardian. The first diagnosed member in each household was defined as the index case. Each household represented a transmission cluster. In total, 258 clusters were included, comprising 650 individuals ranging in age from 0 to 89 years. All patients were unvaccinated, given the early stages of the pandemic during which the study was conducted. Not all participants consented to serological testing. Fig 1 provides an overview of participant inclusion.

Participants aged ≥18 years were classified as adults, and those <18 years were classified as pediatric.

During the study period (March–June 2020), formal infection control guidance was still evolving in Gran Canaria. Index cases were advised to isolate in a separate bedroom and bathroom when feasible, but adherence to these recommendations was not systematically assessed. As a proxy for close contact and potential failure to isolate, we recorded whether household members shared a bedroom with the index case. School and daycare closures had not yet been implemented at the time of data collection, and children continued attending educational institutions under normal conditions.

## Data and sample collection

Blood, saliva, and epidemiological survey data were collected in July 2020 at the Complejo Hospitalario Universitario Insular Materno-Infantil (CHUIMI) in Las Palmas de Gran Canaria. The study was approved by the local ethics committee.

Data collected included demographic variables (age, sex), household characteristics (number of cohabitants, house size, bedroom sharing (defined as two or more individuals sleeping in the same room, regardless of whether they shared

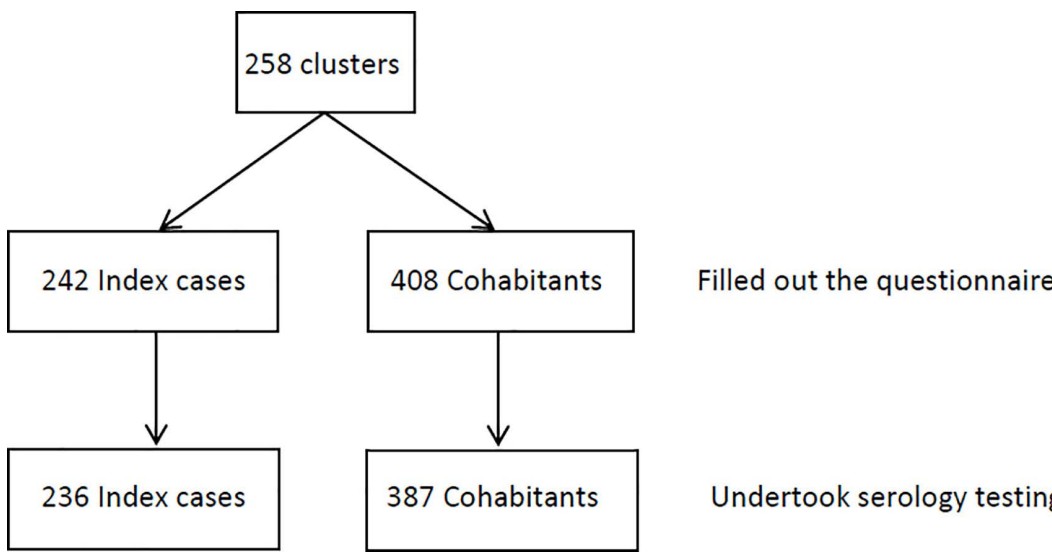

**Fig 1. Summary of study participants.**

a bed)), clinical comorbidities, body weight, body mass index (BMI), and smoking status. Socio-economic status was not evaluated in the study.

## RT-PCR

Molecular confirmation of SARS-CoV-2 infection was performed using real-time RT-PCR assays on nasopharyngeal swabs samples following the kit manufacturers' instructions. The targeted specific genes for SARS-CoV-2 included the nucleocapsid N gene, the RdRP gene of RNA polymerase (Allplex® 2019-nCoV Assay, Seegene, South Korea or Light-Mix® Modular SARS-CoV-2, Tib Molbiol, Germany), and the ORF1a/ b fragment (cobas® SARS-CoV-2 test, Roche, Germany) encoding non-structural proteins for replication. The E gene for the envelope, typical of the Sarbecovirus subgenus, was used as a pan-Sarbecovirus marker.

## Serum IgG to SARS-CoV-2

Serum IgG antibodies against SARS-CoV-2 were assessed by the Microbiology Departments of Dr. Negrín University Hospital and CHUIMI Hospital. Initial testing employed an automated chemiluminescent microparticle immunoassay (SARS-CoV-2 IgG, ARCHITECT, Abbott Diagnostics) targeting nucleocapsid proteins. An index ≥1.4 was interpreted as positive per manufacturer guidelines.

To improve sensitivity and recover potential anti-spike–positive cases missed on initial testing, samples with equivocal results (index 0.1–1.39) were further analyzed using a second chemiluminescent immunoassay (COVID-19 VIRCLIA IgG Monotest, Vircell), targeting both spike and nucleocapsid proteins. An index >1.6 was considered positive.

## ACE2 and TMPRSS2 serine protease expression

Total RNA was extracted from cells obtained from participants' saliva samples using TRIzol (Invitrogen), and complementary DNA (cDNA) was synthesized using the iScript kit (Bio-Rad). As a quality control measure, nasopharyngeal and saliva swabs from healthy controls (n = 14) yielded RNA of similar quality and quantity. *ACE2* fragment amplification was performed via RT-PCR analysis, following methods described by Burgueno et al. in intestinal tissues [21], with minor modifications for use in saliva samples. *TMPRSS2* expression was evaluated using primer sequences and cycling conditions adapted from Ma et al. in ocular surface tissues [22]. Due to the very low expression of the ACE2 gene in saliva, a pre-amplification step was performed for ACE2 prior to quantification. Relative *TMPRSS2* and *ACE2* gene expression levels were calculated using the standard curve method. Reference standard curves were derived from pooled RNA obtained from healthy controls. β-actin was used as the housekeeping gene for normalization.

## Statistical analysis

All statistical analyses were performed using R version 4.0.2 (R Core Team, 2020). Quantitative variables were summarized as means, standard deviations, medians, and interquartile ranges. Categorical variables were presented as frequencies and percentages.

The Kolmogorov–Smirnov test was used to assess the normality of distributions. For group comparisons, the Student's t-test or Mann–Whitney U test was used depending on normality and sample size. Associations between categorical variables were evaluated using Fisher's exact test. Logistic regression models were constructed to predict dichotomous outcomes. Model performance was assessed using the area under the receiver operating characteristic (ROC) curve. A two-sided p-value < 0.05 was considered statistically significant.

## Ethics approval and consent to participate:

The study was conducted according to the guidelines of the Declaration of Helsinki, and approved by the Ethics Committee of Las Palmas University Hospital Dr. Negrín (protocol code 2020-253-1 COVID-19, approved on June 26th 2020).

Written informed consent was obtained from all participants. For minors, consent was obtained from a parent or legal guardian.

## Results

### Demographics

Among the 242 index cases, 239 were adults and only 3 were under 18 years of age (aged 1, 12, and 15 years). The majority of household members were also adults, with only 89 individuals younger than 18 years. Table 1 summarizes the participants' characteristics.

### Source of infection

The most frequently reported sources of infection were household transmission (38%) and travel to high-incidence areas (30%). Table 2 summarizes all sources of infection.

### Serologic results

Of 623 participants who underwent serological testing, 315 (50.5%) tested positive for anti-nucleocapsid IgG antibodies. Among 103 participants who tested negative for anti-nucleocapsid IgG, 68 tested positive for anti-spike IgG, resulting in an adjusted seroprevalence of 61.4%. Among household contacts (excluding index cases), seroprevalence was 42%. Overall, 383/623 participants tested positive for IgG antibodies against SARS-CoV-2.

**Table 1. Participants' characteristics.**

|  |  | Index patients | Cohabitants |
|---|---|---|---|
| Number of clusters | 258 |  |  |
| Family members per household (mean (SD)) | 2.6 (1.3) |  |  |
| Clusters with children < 18 years | 67 |  |  |
| Total participants | 650 | 242* | 408 |
| Mean age (years (SD)) | 44.7 (21) | 54 | 39.5 |
| Group age < 18 years (N (%)) | 89 (13.7%) | 3(1.2%) | 86(21%) |
| Mean age (years (SD) | 8.8 (5.3) | 9.3 (7.3) | 8.8 (5.3) |
| Age range (years) | 0-17 | 1-15 | 0-17 |
| Group age > 18 years (N (%)) | 561 (86.3%) | 239(98.8%) | 322/79%) |
| Mean age (years (SD) | 50.3 (16.4) | 54 | 47 |
| Age range | 18-89 y | 20-87 | 18-89 |
| 18-49 years | 271 | 99 | 171 |
| 50-65 years | 163 | 71 | 92 |
| ≥65 years | 128 | 69 | 59 |
| Female participants (%) | 53.7 | 56 | 52.5 |

*Some index cases did not participate in the study, but their household members did.

**Table 2. Sources of infection.**

| Source of infection (N = X) | Health worker | Essential worker | Travel | Cohabitants | Party | Other sources* |
|---|---|---|---|---|---|---|
| % | 5.8 | 11.2 | 30.1 | 38 | 0.2 | 14.7 |

*Unspecified other sources of infection.

In adults, seropositive individuals were significantly older than seronegative individuals (mean age 53.5 vs. 44.7 years; p<0.001). Conversely, in children, seropositive individuals were significantly younger than seronegative children (mean age 7 vs. 10 years; p=0.038) (Fig 2).

When examining the correlation between age and the antibody indexes, among adults, nucleocapsid antibody index correlated positively with age (Pearson's r=0.44, p<0.001) (Fig 3), but no significant correlation was found for anti-spike antibodies (Pearson's r=0.4; p=0.89), regardless of disease severity. In children, we observed a non-significant negative correlation between age and nucleocapsid antibody index (Pearson's r=−0.36, p=0.11) (Fig 3).

## Clinical characteristics

Symptoms were reported in 83.6% of IgG-positive participants and 28% of IgG-negative participants. Asymptomatic infection was significantly less common in adults than in children (15% vs. 44.4%; OR = 0.21, 95% CI: 0.08–0.53; p<0.002).

In adults, the most common symptoms were fever, cough, and anosmia. Among children, cough and odynophagia were most frequent. Additional symptoms, including asthenia and headache, were categorized under "other symptoms" (Table 3).

Adults experienced more severe disease. No children in the sample developed pneumonia or required hospitalization. In contrast, among adults, 77 developed pneumonia, 96 were hospitalized, 24 required intensive care, and 38 died — all aged >61 years.

## Transmission dynamics

SARS-CoV-2 transmission within the household occurred in 54.4% of participating families. In families with children, 7.6% had at least one child who tested positive for SARS-CoV-2 IgG.

Risk factors for household transmission included older age (p=0.02; OR = 1.05; 95% CI: 1.02–1.08) and sharing a bedroom with the index case (p<0.01; OR = 6.8; 95% CI: 2.8–17.6). No significant association was found with sex, comorbidities, BMI, smoking, or house size.

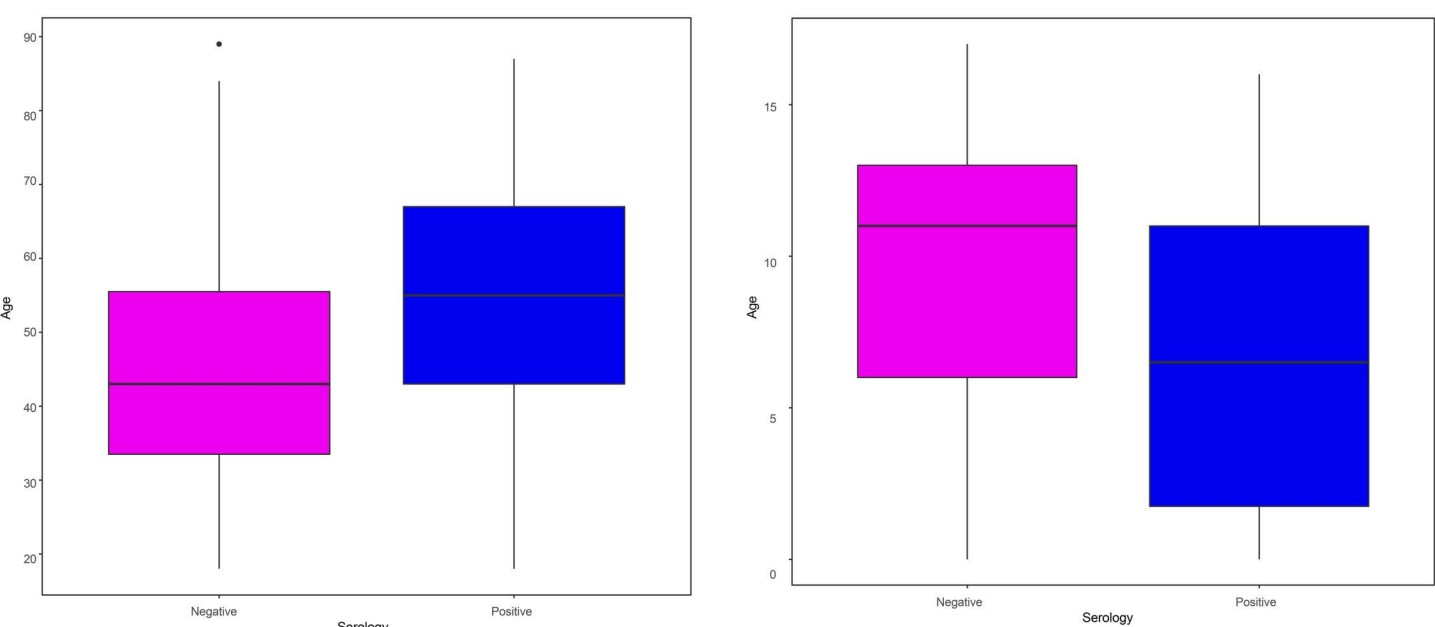

**Fig 2. Age difference between patients with negative and positive serologic results in adults(a) and children(b).**

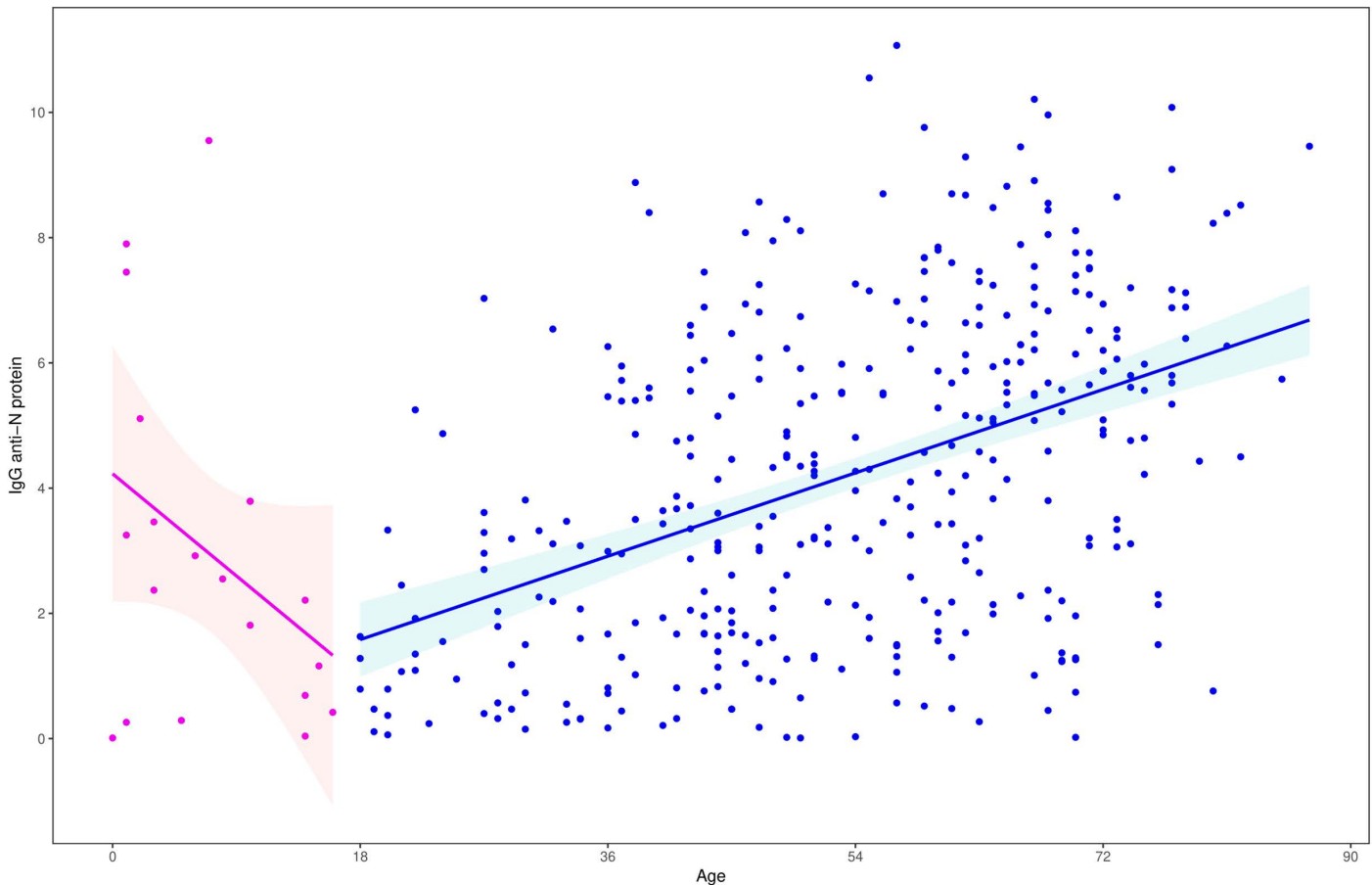

**Fig 3. Correlation between age and antibody indexes in both children and adults.**

**Table 3. Most common symptoms and frequency of appearance among IgG-positive patients.**

| Symptom | Frequency (global) | Frequency (adults) | Frequency (children) | OR (p-value) |
|---|---|---|---|---|
| Fever | 57,2% | 58,1% | 38,8% | 2.5 (0.84, 4.8) |
| Cough* | 54.8% | 56,2% | 27,7% | 3.77 (1.34, 8.8)* |
| Anosmia* | 54.8% | 57,3% | 5,6% | 25.2 (3.50, 227.4)* |
| Myalgia* | 52,7% | 54,8% | 11,1% | 10.6 (2.61, 56.7)* |
| Diarrhea* | 42% | 43,6% | 11,1% | 4.2 (1.59, 37.2)* |
| Nasal discharge | 40,2% | 41,1% | 22,2% | 2.8 (0.77, 7.2) |
| Difficulty breathing* | 36% | 37,8% | 0% | NE* |
| Odynophagia | 32,6% | 33,2% | 22,2% | 1.9 (0.51, 5.6) |
| Abdominal pain | 22,1% | 22,7% | 11,1% | 1.7 (0.60, 14.3) |
| Cutaneous manifestations | 21,5% | 17,8% | 16,6% | 1.1 (0.26, 3.6) |
| Vomiting | 12,5% | 13,2% | 0% | NE |
| Asymptomatic | 16.4% | 15% | 44.4% | 0.2 (0.08, 0.53) |

*These symptoms presented significantly more in adults than in children; NE: not estimable.

The secondary attack rate among household contacts showed that children had significantly lower risk of acquiring infection compared to adults (21.7% vs. 46%; p = 0.002). Table 4 summarizes household transmission data. Due to the limited number of pediatric index cases, we could not evaluate transmission from children to others.

**ACE2 and TMPRSS2 expression**

We obtained valid samples for analysis of mRNA expression from 488 patients for *ACE2* and 576 patients for *TMPRSS2*. Sample characteristics are listed in Table 5.

ACE2 expression showed a weak but statistically significant inverse correlation with age in the total population (Spearman's r = −0.11, p = 0.02). In subgroup analysis, this correlation was significant in children (r = −0.34, p = 0.035), but not in adults (r = −0.03, p = 0.53) (Fig 4a1, b1). However, after applying the Bonferroni correction for multiple comparisons (three tests), the adjusted p-values were as follows: total population p = 0.06, children p = 0.105, and adults p = 1.00. Thus, none of the correlations remained statistically significant after adjustment.

ACE2 expression was not significantly associated with sex, BMI, smoking, antibody index, seropositivity, symptom severity, hospitalization, ICU admission, or underlying disease (Table 6). However, expression was lower in individuals who experienced diarrhea (median expression: 158 vs. 212; p = 0.025).

Regarding *TMPRSS2* expression, we found no significant correlation with age (Fig 4a2, b2), BMI, or antibody indexes. We found no differences in expression when categorizing by sex, smoking, positive serologic results, the severity of symptoms (including hospitalization and ICU care), or previous disease (Table 6).

## Discussion

In this study, we examined the dynamics of household transmission of SARS-CoV-2, with a particular focus on age-related differences in seroprevalence and disease severity, as well as the potential contribution of ACE2 and TMPRSS2 expression to infection susceptibility. Our findings provide insight into pediatric vulnerability, the utility of combined serologic testing, and host-related mechanisms of viral entry in the context of an early-pandemic island population.

**Serologic detection strategy and household transmission dynamics**

By combining two serological assays, using sequential nucleocapsid and spike-based assays, we improved detection by 10%, yielding a seroprevalence of 61.4% substantially higher than the 23.1% reported in the nationwide ENE-COVID

**Table 4. Risk of acquiring SARS-CoV-2 infection in the household (children vs. adults).**

|  | Positive serology | Negative serology | Total |
|---|---|---|---|
| Children (not index case) | 19 | 55 | 74 |
| Adults (not index case) | 144 | 169 | 313 |
| Total | 163 | 224 | 387 |

OR: 0.56 (95% CI: 0.37-0.84); p = 0.002.

**Table 5. Characteristics of patients with *ACE2ACE2* and *TMPRSS2* expression.**

|  | Adults (N) | Children (N) | Total (N) | Quantification (mean (SD); median) |  | Adults (N) | Children (N) | Total (N) | Quantification (mean (SD); median) |
|---|---|---|---|---|---|---|---|---|---|
| **ACE2** | 444 | 44 | **488** | 198.9 (537.8); 110.2 | **TMPRSS2** | 518 | 58 | **576** | 166.76 (528); 93.3 |
| **IgG positive** | 271 | 10 | 281 | 174.2 (256.2); 109.6 | **IgG positive** | 313 | 14 | 327 | 171.94 (583.9); 93.3 |
| **IgG negative** | 173 | 34 | 207 | 232.6 (797.4); 117.7 | **IgG negative** | 205 | 44 | 249 | 155.5 (484.8); 94.2 |

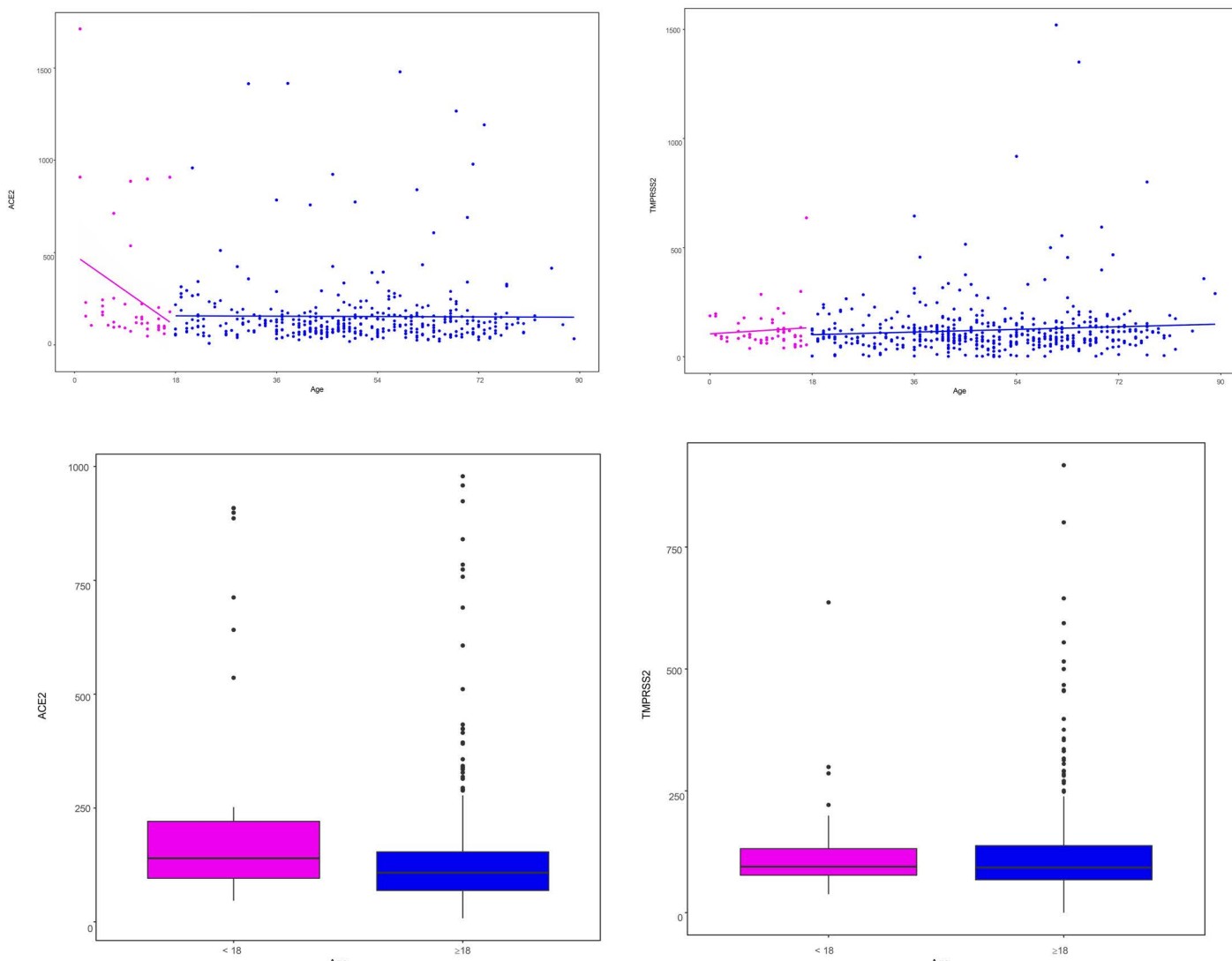

**Fig 4. Correlation between age and ACE2ACE2 and TMPRSS2 protease expression.** (a) Correlation between age and ACE2ACE2 and TMPRSS2 expression. (b) Comparison between ACE2 and TMPRSS2 expression in children and adults.

**Table 6. Correlation and median comparison. ACE2 and TMPRSS2 expression.**

| | BMI[a] | Antibody indexes[a] | Sex[b] (male vs. female) | Smoking[b] (yes vs. no) | IgG anti-N protein[b] (positive vs. negative) | IgG anti-S protein[b] (positive vs. negative) | Hospitalization[b] (yes vs. no) | Hospitalized in ICU[b] (yes vs. no) | Any symptoms[b] (yes vs. no) |
|---|---|---|---|---|---|---|---|---|---|
| **TMPRSS2** | r=0.01; p=0.88 | r=0.02; p=0.66 | 100 vs. 90; p=0.128 | 94 vs. 110; p=0.54 | 95 vs. 92; p=0.6 | 88 vs. 104; p=0.15 | 103 vs. 92; p=0.16 | 125 vs. 97; p=0.12 | 95 vs 92; p=0.44 |
| **ACE2** | r=−0.03; p=0.6 | r=−0.07; p=0.16 | 114 vs. 109; p=0.948 | 112 vs. 110, p=0.9 | 108 vs. 117; p=0.11 | 116 vs. 112; p=0.98 | 107 vs. 113; p=0.38 | 93 vs. 111; p=0.98 | 109 vs 113; p=0.27 |

a)Correlation; Spearman's r; p-value; b) Mann–Whitney U. Comparison among medians.

study by Pollán et al. for Spain [23]. This discrepancy likely reflects our non-random sampling focused on intra-household exposure and participation bias, as families concerned about possible transmission were more likely to volunteer.

Secondary attack rates in household settings vary widely. While our observed rates align with those of Maltezou et al. (60%) [24] and Grijalva et al (53%) [25], they are notably higher than those reported in studies from Wuhan (16% [10], − 30% [26]) and the meta-analysis by Madewell et al. [27](16.6%). This variability may reflect differences in testing strategies, as many studies relied solely on RT-PCR and may have understimated true exposure [28]. Even though our attack rates differ, we report similar results regarding the effect of age and co-sleeping with index cases.

### Age-related susceptibility and disease severity

We found a decreased seroprevalence in children compared to adults (21.7% vs. 46%), with a significantly lower secondary attack rate (OR: 0.56, 95% CI: 0.37–0.84), consistent with the results from the meta-analysis by Viner et al. (OR: 0.56) [29]. Published studies have shown conflicting results [9,30]. It is important to note that these earlier findings were based on infections with the original SARS-CoV-2 strains. Emerging evidence from later pandemic waves suggests that pediatric transmission may increase under variant-specific conditions [31,32], with higher infection rates and more symptomatic presentations reported among children during the circulation of the Delta and Omicron variants [16].

In our sample, the effect of age on disease susceptibility and severity was observed in both children and adults, with children presenting milder forms of the disease, as has been described in the literature [33]. During the study period, none of the approximately 106,672 children under 14 years of age living in Gran Canaria [34] presented with pneumonia or required hospitalization due to SARS-CoV-2 infection.

Regarding antibody indexes in children, Torres et al. [35] reported an association between younger age and positivity rates. We did not find such an association; however, we found a positive correlation between antibody indexes and age, with children presenting lower indexes than adults, and severity, with symptomatic and hospitalized patients presenting higher antibody indexes than asymptomatic patients. These results are aligned with publications from other groups [14,36].

The mechanistic basis for lower susceptibility and disease severity in children remains uncertain, as well as the role played by an increased or decreased antibody index in SARS-CoV-2 infection. It has been hypothesized [37] that the infection is less severe in children due to an attenuated immune response, possibly resulting in viral tolerance [38]. Weisberg et al. [39] suggested that differences in the immune response between children and adults, with an increased anti-S(anti-spike) response in children, might provide protection in the younger age group. The decreased severity was also proposed to result from trained innate immunity conferred by live attenuated vaccines such as measles, mumps, and rubella [40]. Furthermore, ageing per se is associated with IFN-1 dysfunction, and immunity mediated by plasmacytoid dendritic cells decreases with age, possibly contributing to increased disease severity in adults [41].

### *ACE2* and *TMPRSS2* expression

ACE2 has garnered significant attention as a potential determinant of SARS-CoV-2 infectivity since its role in viral cell entry was first established. Initial studies showed reduced *ACE2* expression in the upper respiratory tract of children, suggesting that lower receptor availability may serve as a protective factor against SARS-CoV-2 infection [4,6,42,43]. These early studies were performed in healthy individuals and did not include infected children. Yonker et al. [14] reported increased expression in children older than 10 years and in those who had been infected by SARS-CoV-2. In contrast, in our series we found a different gradient of expression, with slightly increased expression of *ACE2* in children <18 years of age compared with adults, and increased expression in younger compared with older children. However, after applying the Bonferroni correction for multiple comparisons, these associations no longer reached statistical significance, indicating that they should be interpreted with caution and considered exploratory and hypothesis-generating rather than confirmatory. As such, our findings underscore the need for further research in larger, independent cohorts to clarify the relationship between *ACE2* expression and age.

We found no significant differences in *ACE2* expression related to seropositivity, sex, BMI, smoking status, or severity of the disease. ACE2 is located on the X chromosome, and although X-linked genes may show sex-specific expression patterns, we did not observe significant differences in ACE2 expression between males and females in our cohort. These findings suggest no overt sex-based regulation in saliva, although differential expression in other tissues or through mechanisms such as X-inactivation escape cannot be excluded.

Given the close interaction between ACE2 and other host cell proteases such as TMPRSS2 [3] and HAT [44], we decided to analyze *TMPRSS2* expression. We found no significant differences in expression with age, BMI, sex, infection, or severity of the disease. Thus, our findings do not support a relationship between TMPRSS2 expression levels and COVID-19 severity. We also examined the effect of comorbidities on ACE2 and TMPRSS2 expression. Due to small numbers in individual subgroups, comorbid conditions, including diabetes and asthma, were analyzed as a composite variable. This aggregate measure was not significantly associated with gene expression levels. However, given prior evidence linking diabetes and asthma to modified ACE2 expression [45], further studies with larger, stratified cohorts are needed to evaluate condition-specific effects more precisely. Emerging evidence also suggests that respiratory tract colonization by Lactobacilli may modulate ACE2 expression and differs across age groups [46]. Although we did not assess colonization status, this represents another relevant factor that should be explored in future studies.

Conflicting findings complicate establishing the role of the quantitative expression of both ACE2 and TMPRSS2 in the susceptibility to and severity of SARS-CoV-2 infection. These discrepancies may reflect tissue-specific variability, including differential expression of *ACE2* and *TMPRSS2* in the nasopharynx, oral mucosa or lower airway epithelium,. However, perhaps it is not solely the amount of receptor and protease expressed in the cell's surface, but rather their functionality, which is probably linked to genetic polymorphisms, which account for the differential effect of the virus in the population [47–49]. This is supported by studies identifying ACE2 polymorphisms associated with altered binding efficiency and susceptibility [50,51]. Our study focused on gene expression and did not include genotyping or sequencing analyses. Future studies integrating both gene expression and host genetic profiling will be important to better understand the interplay between receptor quantity, structure, and COVID-19 susceptibility.

The strengths of our study are: (1) the use of serology testing in both symptomatic and asymptomatic household members; (2) the combined use of two distinct antibody assays to improve diagnostic sensitivity; (3) Unique context of an island population under early lockdown, limiting external transmission and enhancing the accuracy of household transmission attribution.

An important limitation of our study lies in the statistical power of certain subgroup analyses. Although the overall sample size was relatively large, the number of pediatric participants with valid ACE2 and TMPRSS2 expression data was limited. Post hoc power analysis revealed that the comparisons of ACE2 and TMPRSS2 expression between children and adults yielded small effect sizes (Cohen's d = 0.18 and 0.15, respectively), resulting in low statistical power (<25%). These results suggest that the analysis may have been underpowered to detect subtle differences in receptor expression between age groups. In contrast, the comparison of seropositivity rates between children and adults demonstrated a large effect size (Cohen's w = 2.42) and excellent power (>99%). Similarly, the association between age and seropositivity in adults (Cohen's d = 0.42; power = 96%) reinforces the robustness of the age gradient observed in this group. However, the corresponding analysis in children, despite a larger effect size (Cohen's d = 0.57), was only moderately powered (56%), limiting the strength of inference that can be drawn.

Additional limitations relate to study design and contextual factors during the early phases of the pandemic. At that time, RT-PCR testing was limited to individuals with a clear epidemiological link and typical symptoms (fever and respiratory signs), resulting in non-random sampling and a clear selection bias. As a result: (1) asymptomatic individuals and those without a known exposure were likely underdiagnosed; (2) only around half of the target population agreed to participate, introducing potential participation bias; (3) index case identification was based on reported symptom onset, which may not accurately reflect the true transmission chronology; (4) symptom recall was subject to bias due to the lag between the

symptomatic period and the survey (1–4 months); and (5) secondary infections may not have occurred exclusively within the household. Furthermore, it is likely that families with symptomatic individuals or known exposures were more motivated to participate, potentially leading to an overrepresentation of higher-risk households. This selection bias limits the generalizability of our findings to the broader community. It is also important to acknowledge that the early-pandemic context and the geographic isolation of the study population may limit extrapolation to other regions or later stages of the pandemic.

Additionally, since our study was conducted prior to the emergence of major variants such as Delta and Omicron, during a period dominated by the ancestral Wuhan strain, we cannot exclude the possibility that variant-specific differences in transmissibility and susceptibility could alter the patterns observed.

## Conclusions

In this study, we demonstrate that a combined serologic strategy enhances detection of past SARS-CoV-2 infections in household settings. Although younger children showed slightly higher ACE2 expression, no statistically significant correlations with age, susceptibility, or disease severity were found after multiple testing correction. Children were less likely to acquire infection and generally experienced milder disease. Transmission risk was influenced by age and household factors such as co-sleeping with an index case. These findings reinforce the lower risk of infection and severe illness in children and suggest that ACE2/TMPRSS2 expression alone does not explain these differences. Given the absence of significant associations and the study's limitations, further research is needed to elucidate the biological mechanisms underlying age-related differences in SARS-CoV-2 susceptibility and outcomes.

## Author contributions

**Conceptualization:** Yeray Nóvoa-Medina, Abián Montesdeoca-Melián, Araceli Hernández-Betancor, Francisco J. Rodríguez-Esparragón, Svetlana Pavlovic-Nesic, Melisa Hernández-Febles, Jesús M. González-Martín, Martín Castillo De Vera, Carlos Rodríguez-Gallego, Luis Peña-Quintana.

**Data curation:** Yeray Nóvoa-Medina, Francisco J. Rodríguez-Esparragón.

**Formal analysis:** Yeray Nóvoa-Medina, Francisco J. Rodríguez-Esparragón, Melisa Hernández-Febles, Jesús M. González-Martín, Laura Cappiello.

**Funding acquisition:** Yeray Nóvoa-Medina.

**Investigation:** Jesús Poch-Páez, Yeray Nóvoa-Medina, Abián Montesdeoca-Melián, Araceli Hernández-Betancor, Francisco J. Rodríguez-Esparragón, Svetlana Pavlovic-Nesic, Melisa Hernández-Febles, Laura Cappiello, Valewska Wallis-Gómez, Joaquin Quiralte-Castillo, Alejandro Maj10-Rodríguez, Martín Castillo De Vera, Maria T. Angulo-Moreno, Augusto González-Pérez, Asunción Rodríguez, Zelidety Espinel-Padrón, Elisa M. Canino-Calderín, Irina Manzano-Gracia, Elena Colino-Gil, Ana I. Reyes Dominguez, Irina Moreno-Afonso, Raquel McLaughlin-García, Maria L. Naranjo-Báez, Ana Bordes-Benitez, Isabel De Miguel-Martínez, Carlos Rodríguez-Gallego, Luis Peña-Quintana.

**Methodology:** Yeray Nóvoa-Medina, Abián Montesdeoca-Melián, Araceli Hernández-Betancor, Francisco J. Rodríguez-Esparragón, Melisa Hernández-Febles, Jesús M. González-Martín, Martín Castillo De Vera, Ana Bordes-Benitez, Carlos Rodríguez-Gallego, Luis Peña-Quintana.

**Project administration:** Jesús Poch-Páez, Yeray Nóvoa-Medina.

**Resources:** Yeray Nóvoa-Medina, Francisco J. Rodríguez-Esparragón.

**Supervision:** Yeray Nóvoa-Medina.

**Visualization:** Jesús M. González-Martín, Joaquin Quiralte-Castillo.

**Writing – original draft:** Jesús Poch-Páez, yeray nóvoa-medina, Abián Montesdeoca-Melián, Francisco J. Rodríguez-Esparragón, Melisa Hernández-Febles, Jesús M. González-Martín.

**Writing – review & editing:** Jesús Poch-Páez, Yeray Nóvoa-Medina, Abián Montesdeoca-Melián, Araceli Hernández-Betancor, Francisco J. Rodríguez-Esparragón, Svetlana Pavlovic-Nesic, Melisa Hernández-Febles, Jesús M. González-Martín, Laura Cappiello, Valewska Wallis-Gómez, Joaquin Quiralte-Castillo, Alejandro Majan-Rodríguez, Martín Castillo De Vera, Maria T. Angulo-Moreno, Augusto González-Pérez, Asunción Rodríguez, Zelidety Espinel-Padrón, Elisa M. Canino-Calderín, Irina Manzano-Gracia, Elena Colino-Gil, Ana I. Reyes Dominguez, Irina Moreno-Afonso, Raquel McLaughlin-García, Maria L. Naranjo-Báez, Ana Bordes-Benitez, Isabel De Miguel-Martínez, Carlos Rodríguez-Gallego, Luis Peña-Quintana.

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
