## [Decision Letter · Decision Letter 0]

Dear Dr. nóvoa-medina,

Thank you for submitting your manuscript to PLOS ONE. After careful consideration, we feel that it has merit but does not fully meet PLOS ONE’s publication criteria as it currently stands. Therefore, we invite you to submit a revised version of the manuscript that addresses the points raised during the review process.

**ACADEMIC EDITOR: **

**Overall Assessment:**

This manuscript presents a population-based, cross-sectional analysis evaluating ACE2 and TMPRSS2 expression in relation to SARS-CoV-2 susceptibility and transmission within households. The study is timely and methodologically sound, with detailed serological and molecular assessments. It adds meaningful insights to the understanding of pediatric susceptibility and receptor biology in COVID-19.

However, a few revisions are required to enhance transparency, clarify methodology, and support key conclusions. Below is a structured editorial assessment:

<h3 data-end="1551" data-start="1480">**Methods and Statistical Analyses** .</h3>

**Concerns:****Sample size calculation** was not provided, which is critical for assessing power and validity. Please include or justify its absence.Clarify handling of potential **confounding variables** , especially in logistic regression models for transmission risk (e.g., socio-economic status, comorbidities, vaccination status, if relevant).Discuss the implications of **selection bias** more thoroughly—families with symptomatic individuals may have been more likely to participate.<h3 data-end="2689" data-start="2649">**Sample Size Calculations ** </h3>**Required Action:** Please include a justification or post hoc power analysis to support the adequacy of the sample size, particularly for subgroup analyses in children.<h3 data-end="3564" data-start="3502">** Limitations Discussion ** </h3>**Suggestions:**Discuss the **generalizability** of results, given the unique context of early-pandemic Gran Canaria.Clarify that **variant influence** cannot be excluded due to the early sampling period.<h3 data-end="3947" data-start="3902">**Additional Recommendations:** </h3>**Title:** Consider shortening for clarity. Suggested: “Age-dependent ACE2/TMPRSS2 expression and SARS-CoV-2 household transmission in Gran Canaria.”**Data Availability:** Currently states data available on request. PLOS encourages open data. Please consider uploading de-identified datasets to a public repository if feasible, or strengthen the rationale for restrictions.

We look forward to receiving your revised manuscript.

Kind regards,

Mohamed Samy Abousenna

Academic Editor

PLOS ONE

Journal Requirements:

5. In the online submission form, you indicated that the datasets generated and/or analyzed during the study are not publicly available due to participant confidentiality, but are available from the corresponding author upon reasonable request.

6. Please remove all personal information, ensure that the data shared are in accordance with participant consent, and re-upload a fully anonymized data set.

Reviewers' comments:

Reviewer's Responses to Questions

**Comments to the Author**

1. Is the manuscript technically sound, and do the data support the conclusions?

Reviewer #1: Yes

Reviewer #2: Yes

Reviewer #3: Partly

Reviewer #4: Yes

Reviewer #5: Yes

2. Has the statistical analysis been performed appropriately and rigorously?

Reviewer #1: Yes

Reviewer #2: Yes

Reviewer #3: Yes

Reviewer #4: Yes

Reviewer #5: Yes

3. Have the authors made all data underlying the findings in their manuscript fully available?

Reviewer #1: Yes

Reviewer #2: Yes

Reviewer #3: No

Reviewer #4: No

Reviewer #5: No

4. Is the manuscript presented in an intelligible fashion and written in standard English?

Reviewer #1: Yes

Reviewer #2: Yes

Reviewer #3: Yes

Reviewer #4: Yes

Reviewer #5: Yes

Reviewer #1: The authors aimed to assess whether the expression of ACE2 and TMPRSS2 is associated with susceptibility to and severity of COVID-19 across age groups. The study is well designed but vaccination status of the participant did not mentioned. So authors should mention it. Also which variant was observed duration of the study in the region? ACE2 is located on X chromosome so males are hemizygous for the ACE2 gene. Males and females might have different levels of ACE gene expression. This information and expression results for both gender should be discussed. Why the authors did not check the ACE2 and TMPRSS2 gene variants that also effect the receptor structure? If they can also discuss it it will give more scientific value to the manuscript.

Reviewer #2: Age-related ACE2 and TMPRSS2 expression and household transmission of SARSCoV-

2:Insights from a population-based study in Gran Canaria

After reviewing the research and its findings, I find that the study is well-conducted and its results are accurate and valid. Therefore, I recommend its acceptance in its current form

Reviewer #3: Poch-Pàez and colleagues describe a cross-sectional study conducted in Gran Canaria between March 10, 2020 and June 2, 2020 evaluating age-related ACE2 and TMPRSS2 gene expression (saliva samples) and household transmission of SARS-CoV2. Their data encompass 258 households with 650 individuals (of whom 89 were children <18 years of age). Their findings did not support a strong relationship between expression of these 2 genes and susceptibility or severity of COVID-19 and noted that children appeared to be less susceptible to SARS-COV-2 infection and to have milder disease.

The study is interesting, although somewhat confusing in places. Overall, it’s value currently is diminished by the many other published studies on transmission dynamics, the emergence of many newer variants with differing infectivities and severities of infection, the introduction of COVID-19 vaccinations, and the shrinking pool of SARS-CoV-2 naïve adults and children.

I have some questions and concerns that the authors should address:

1) The study was conducted early in the pandemic (March to June 2020). Their statement in the Background section that “published data point toward similar (7) or even lower (8,9) infection compared to adults” when referring to pediatric SARS-CoV-2 infection uses references published in 2020 (again, early in the pandemic) and does not consider how infectivity of other SARS-CoV-2 variants differ from the initial clades. For example, in a CDC study that compared seroprevalence rates between September-December 2021 to January-February 2022 (emergence of Omicron B.1.1.529 variant), seroprevalence among children 0-11 years of age increased from 44.2% to 75.2%, and from 45.6% to 74.2% among those aged 12-17 years, respectively. Among adults ages 18-49 years, seroprevalence increased from 36.5% to 63.7%, and from 28.8% to 49.8% among those 50-64 years of age. The CDC study showed that children had higher SARS-CoV-2 seroprevalence than adults. This difference from the current authors’ results is likely due to the infectivity of circulating SARS-CoV-2 variants at the time of the study, and probably geographic differences. [Clark et al. MMWR Morb Mortal Wkly Rep 2022;71:606-8].

2) In the Methods-Data and Sample Collection, did the authors collect data on socioeconomic status of the 258 household “clusters”? Higher socioeconomic status is generally associated with lower SARS-CoV-2 antibody prevalence [Naeimi et al. eClinical Medicine 2023;56:101786]

3) Similarly, do we know how many individuals were diabetic, since the condition is associated with higher ACE2 expression in many tissues (lungs, kidneys, pancreas, etc)? Conversely, patients with allergic sensitization and asthma have lower ACE2 gene expression [Shukla. Eur Arch Otorhinolaryngol 2020;278:2637-40].

4) Lactobacilli colonizing the respiratory tract can down-regulate ACE2 expression. Rates of colonization in infants, children, and adults vary. This was not examined as a possible variable. [Taufer and Rampeletto. Microorganisms 2024;12:284]

5) Is “sharing a bedroom” the same as sharing a bed (Transmission Dynamics section)?

6) In Table 5, it would be helpful to show the amount of ACE2 and TMPRSS2 gene expression between those who are IgG positive and those who are not.

7) In the title of Table 5, please change “ECA2” to “ACE2”. Also, the same error appears in Table 6, row 3.

8) Were schools/day cares closed in Gran Canaria during the study period, as this could provide another explanation for the lower seropositivity in children. What kind of infection control/isolation advice was given at the time to infected index cases (e.g., masking, self-quarantine, staying alone in a room)? Was there any difference in household transmission rates among those who were hospitalized/died (less exposure time to family members) versus those who were not?

9) Studies have revealed genetic polymorphisms of ACE1, ACE2, IFTM3, TMPRSS2 and TNFα genes associated with increased risk of infection and illness severity. We do not know what polymorphism are present in the individuals described in this study, both infected and uninfected. [Möhlendick et al. Pharmacogenet Genomics 2021;31:165-71; Pecoraro et al. Clin Exp Med 2023;23:3251-64]

Reviewer #4: 1. Saliva‑based RT‑PCR quantification of ACE2/TMPRSS2 may not reflect receptor abundance in the nasopharynx or lower airway epithelium, where viral entry occurs. Include a paragraph comparing saliva to nasal/nasopharyngeal sampling, citing relevant literature.

2. "ECA2” appears in Table 6 header instead of “ACE2".

3. “TMPRSS22” instead of “TMPRSS2” in Keywords and several figure legends.

4. Reference formatting: check consistency of journal names (some are abbreviated, some full).

5. Numerous hypothesis tests (Tables 3–6, correlations, subgroup comparisons) are reported at p<0.05 without correction. Justify why unadjusted p‑values are acceptable. Report adjusted p‑values for the most important findings (e.g., ACE2 vs. age correlations).

Reviewer #5: Summary:

This paper examines gene expression as it relates to SARS-CoV-2 household transmission. A cross-sectional study among households was conducted to evaluate gene expression and symptoms among index and household contacts. Although the methodology seems solid, there needs to be more contextualization of the work in this setting and a better sense of the population, target population, and sociodemographic context. The authors could also expand on the implications of this study in terms of how it relates to other island countries and similar populations. The discussion could be condensed to focus more on the main points and how they relate to the literature, with clear sections involving key findings and how they relate to the literature, limitations, and strengths of the study. Also, since this study was done before the Delta and Omicron waves, that should be noted clearly in the limitations section. Also, all of the graphs below should be optimized for individuals with colorblindness, and have clear legends with titles and labeled axes. Finally, This manuscript would benefit from a great deal of copy editing and proofreading to help improve grammatical issues

Specific comments

Page 5 – remove the comma between Some and studies

Page 5 – move the (8,9) citation to after the period after adults

Page 6 – it may be helpful to add some information about the size of Gran Canaria, and the population of children and adults.

Page 11 – be sure to proofread the table, and make sure that parentheses match for the group age row.

Page 12 – you should be specific for your serologic results and indicate if these were among household contacts or the index cases. For example, you could say that, Among household contacts, seroprevalence was 42%

In table 3 on Page 13, you should include the 95% Confidence interval for each odds ratio. If you add that, you will not have to add the p value, and then you can bold or add a * to all of the odds ratios that are statistically significant.

For the symptoms that presented significantly more in adults than in children, you should provide these results in a separate table or in the appendix.

Page 17, all of this information about the sampling and its limitations should be moved to a separate limitations section.

In general, the discussion section is very long and could be streamlined. I would suggest picking three main findings to focus on, and then asses how those findings compare to the literature.

Page 21 – You should add information about how these results cannot be extrapolated to the population as a whole because of the study, and may reflect a sicker population than the population of Gran Canaria as a whole. Also, is there any possibility of waning genetic expression over time given that these tests may have been conducted up to 4 months after initial infection?

Page 21 – conclusion, you should reiterate that your study found these things, and that more research is needed.

Pages 38-42 – all of these graphs should have legends and be optimized for black and white printing or red/green colorblindness.

**Do you want your identity to be public for this peer review?** For information about this choice, including consent withdrawal, please see our Privacy Policy

Reviewer #1: **Yes: ** Mahmut Cerkez Ergoren

Reviewer #2: No

Reviewer #3: **Yes: ** Bishara J. Freij, M.D.

Reviewer #4: **Yes: ** Fernando Namuche

Reviewer #5: No

---

## [Author Response · Author response to Decision Letter 1]

25 Jun 2025

ACADEMIC EDITOR:

Overall Assessment:

This manuscript presents a population-based, cross-sectional analysis evaluating ACE2 and TMPRSS2 expression in relation to SARS-CoV-2 susceptibility and transmission within households. The study is timely and methodologically sound, with detailed serological and molecular assessments. It adds meaningful insights to the understanding of pediatric susceptibility and receptor biology in COVID-19.

However, a few revisions are required to enhance transparency, clarify methodology, and support key conclusions. Below is a structured editorial assessment:

Methods and Statistical Analyses.

• Concerns:

o Sample size calculation was not provided, which is critical for assessing power and validity. Please include or justify its absence.

“No formal sample size calculation was performed for this study. The sample size was determined by the number of participants who agreed to participate during the recruitment period. While we aimed to enroll as many participants as possible to maximize statistical power and minimize the risk of type II error, we acknowledge that the absence of a priori sample size estimation may limit the generalizability and statistical robustness of our findings.” This comment has been added in “Methods” section of the manuscript.

o Clarify handling of potential confounding variables, especially in logistic regression models for transmission risk (e.g., socio-economic status, comorbidities, vaccination status, if relevant).

Socio-economic status was not evaluated in the study. All patients were unvaccinated, given the early stages of the pandemic during which the study was conducted. This comment has been added in “Methods” section of the manuscript.

o Discuss the implications of selection bias more thoroughly—families with symptomatic individuals may have been more likely to participate.

This paragraph has been added to the Discussion (limitation) section: “it is likely that families with symptomatic individuals or known exposures were more motivated to participate, potentially leading to an overrepresentation of higher-risk households. This selection bias limits the generalizability of our findings to the broader community.”

o Sample Size Calculations

Required Action: Please include a justification or post hoc power analysis to support the adequacy of the sample size, particularly for subgroup analyses in children.

The following paragraph has been added in the discussion, when describing the limitations of the study: “An important limitation of our study lies in the statistical power of certain subgroup analyses. Although the overall sample size was relatively large, the number of pediatric participants with valid ACE2 and TMPRSS2 expression data was limited. Post hoc power analysis revealed that the comparisons of ACE2 and TMPRSS2 expression between children and adults yielded small effect sizes (Cohen's d = 0.18 and 0.15, respectively), resulting in low statistical power (<25%). These results suggest that the analysis may have been underpowered to detect subtle differences in receptor expression between age groups. In contrast, the comparison of seropositivity rates between children and adults demonstrated a large effect size (Cohen's w = 2.42) and excellent power (>99%). Similarly, the association between age and seropositivity in adults (Cohen's d = 0.42; power = 96%) reinforces the robustness of the age gradient observed in this group. However, the corresponding analysis in children, despite a larger effect size (Cohen's d = 0.57), was only moderately powered (56%), limiting the strength of inference that can be drawn.”

o Limitations Discussion

Suggestions:

Discuss the generalizability of results, given the unique context of early-pandemic Gran Canaria.

We added this paragraph after the Limitations: “Regarding the generalizability of our findings, it is important to acknowledge that the early-pandemic context and the geographic isolation of the study population may limit extrapolation to other regions or later stages of the pandemic.”

Clarify that variant influence cannot be excluded due to the early sampling period.

We have clarified in the Limitations section that our study was conducted prior to the emergence of major variants such as Delta and Omicron, and that variant-specific effects on transmission and susceptibility cannot be excluded.

Additional Recommendations:

Title: Consider shortening for clarity. Suggested: “Age-dependent ACE2/TMPRSS2 expression and SARS-CoV-2 household transmission in Gran Canaria.” Done

Data Availability: Currently states data available on request. PLOS encourages open data. Please consider uploading de-identified datasets to a public repository if feasible, or strengthen the rationale for restrictions.

The following paragraph has been added to the “availability of data and material statement”: The data contain potentially identifying information from a small and geographically confined population, and public sharing was not included in the informed consent approved by the ethics committee. Therefore, the full dataset cannot be made publicly available. However, de-identified data may be made available to qualified researchers upon reasonable request, subject to approval by the ethics committee of the Complejo Hospitalario Universitario Insular-Materno Infantil and compliance with applicable data protection regulations.

• A rebuttal letter that responds to each point raised by the academic editor and reviewer(s). You should upload this letter as a separate file labeled 'Response to Reviewers'. done

• A marked-up copy of your manuscript that highlights changes made to the original version. You should upload this as a separate file labeled 'Revised Manuscript with Track Changes'. done

• An unmarked version of your revised paper without tracked changes. You should upload this as a separate file labeled 'Manuscript'. done

Journal Requirements:

1. Please ensure that your manuscript meets PLOS ONE's style requirements, including those for file naming. The PLOS ONE style templates can be found at https://journals.plos.org/plosone/s/file?id=wjVg/PLOSOne_formatting_sample_main_body.pdf and https://journals.plos.org/plosone/s/file?id=ba62/PLOSOne_formatting_sample_title_authors_affiliations.pdf Done

2. We note that the grant information you provided in the ‘Funding Information’ and ‘Financial Disclosure’ sections do not match. When you resubmit, please ensure that you provide the correct grant numbers for the awards you received for your study in the ‘Funding Information’ section. Done

3. Your ethics statement should only appear in the Methods section of your manuscript. If your ethics statement is written in any section besides the Methods, please move it to the Methods section and delete it from any other section. Please ensure that your ethics statement is included in your manuscript, as the ethics statement entered into the online submission form will not be published alongside your manuscript. DONE

4. We note that you have indicated that there are restrictions to data sharing for this study. For studies involving human research participant data or other sensitive data, we encourage authors to share de-identified or anonymized data. However, when data cannot be publicly shared for ethical reasons, we allow authors to make their data sets available upon request. For information on unacceptable data access restrictions, please see http://journals.plos.org/plosone/s/data-availability#loc-unacceptable-data-access-restrictions. Addressed in the Declarations section of the manuscript

a) If there are ethical or legal restrictions on sharing a de-identified data set, please explain them in detail (e.g., data contain potentially identifying or sensitive patient information, data are owned by a third-party organization, etc.) and who has imposed them (e.g., a Research Ethics Committee or Institutional Review Board, etc.). Please also provide contact information for a data access committee, ethics committee, or other institutional body to which data requests may be sent. DONE

5. In the online submission form, you indicated that the datasets generated and/or analyzed during the study are not publicly available due to participant confidentiality, but are available from the corresponding author upon reasonable request. All PLOS journals now require all data underlying the findings described in their manuscript to be freely available to other researchers, either 1. In a public repository, 2. Within the manuscript itself, or 3. Uploaded as supplementary information. This policy applies to all data except where public deposition would breach compliance with the protocol approved by your research ethics board. If your data cannot be made publicly available for ethical or legal reasons (e.g., public availability would compromise patient privacy), please explain your reasons on resubmission and your exemption request will be escalated for approval. DONE

Reviewers' comments:

Reviewer #1: The authors aimed to assess whether the expression of ACE2 and TMPRSS2 is associated with susceptibility to and severity of COVID-19 across age groups. The study is well designed but vaccination status of the participant did not mentioned. So authors should mention it. DONE

Also which variant was observed duration of the study in the region? The ancestral Wuhan strain. It has been added both in the methodology and discussion sections

ACE2 is located on X chromosome so males are hemizygous for the ACE2 gene. Males and females might have different levels of ACE gene expression. This information and expression results for both gender should be discussed. We thank the reviewer for this important observation. As correctly noted, ACE2 is located on the X chromosome and sex-linked differences in expression are biologically plausible. In our analysis, we included sex as a variable when examining ACE2 and TMPRSS2 expression. However, no statistically significant differences were found between males and females for either gene (ACE2: median 114 vs. 109; p = 0.948; TMPRSS2: median 100 vs. 90; p = 0.128). These results are detailed in Table 6. While our findings do not support sex-related differential expression in saliva samples, we acknowledge that other tissues or regulatory mechanisms (e.g., X inactivation escape, hormonal regulation) may yield different results and warrant further study. We have clarified this point in the discussion section.

Why the authors did not check the ACE2 and TMPRSS2 gene variants that also effect the receptor structure? If they can also discuss it it will give more scientific value to the manuscript.

We thank the reviewer for raising this important point. Indeed, ACE2 and TMPRSS2 genetic variants may influence receptor structure and viral entry, and several polymorphisms have been associated with differential susceptibility to SARS-CoV-2 infection. However, our study was designed to evaluate gene expression levels in saliva and did not include sequencing or genotyping components. We had briefly acknowledged this in the Discussion and have now expanded that section to clarify that genetic variability in these genes could contribute to host susceptibility independently of expression levels and merits further investigation.

Reviewer #2: Age-related ACE2 and TMPRSS2 expression and household transmission of SARSCoV-2:Insights from a population-based study in Gran Canaria

After reviewing the research and its findings, I find that the study is well-conducted and its results are accurate and valid. Therefore, I recommend its acceptance in its current form. We thank the reviewer for their positive assessment.

Reviewer #3: Poch-Pàez and colleagues describe a cross-sectional study conducted in Gran Canaria between March 10, 2020 and June 2, 2020 evaluating age-related ACE2 and TMPRSS2 gene expression (saliva samples) and household transmission of SARS-CoV2. Their data encompass 258 households with 650 individuals (of whom 89 were children <18 years of age). Their findings did not support a strong relationship between expression of these 2 genes and susceptibility or severity of COVID-19 and noted that children appeared to be less susceptible to SARS-COV-2 infection and to have milder disease.

The study is interesting, although somewhat confusing in places. Overall, it’s value currently is diminished by the many other published studies on transmission dynamics, the emergence of many newer variants with differing infectivities and severities of infection, the introduction of COVID-19 vaccinations, and the shrinking pool of SARS-CoV-2 naïve adults and children.

I have some questions and concerns that the authors should address:

1) The study was conducted early in the pandemic (March to June 2020). Their statement in the Background section that “published data point toward similar (7) or even lower (8,9) infection compared to adults” when referring to pediatric SARS-CoV-2 infection uses references published in 2020 (again, early in the pandemic) and does not consider how infectivity of other SARS-CoV-2 variants differ from the initial clades. For example, in a CDC study that compared seroprevalence rates between September-December 2021 to January-February 2022 (emergence of Omicron B.1.1.529 variant), seroprevalence among children 0-11 years of age increased from 44.2% to 75.2%, and from 45.6% to 74.2% among those aged 12-17 years, respectively. Among adults ages 18-49 years, seroprevalence increased from 36.5% to 63.7%, and from 28.8% to 49.8% among those 50-64 years of age. The CDC study showed that children had higher SARS-CoV-2 seroprevalence than adults. This difference from the current authors’ results is likely due to the infectivity of circulating SARS-CoV-2 variants at the time of the study, and probably geographic differences. [Clark et al. MMWR Morb Mortal Wkly Rep 2022;71:606-8]. Thank you for your suggestion. We tried to reflect this by citing Khemiri et als work (Khemiri H, Ayouni K, Triki H, Haddad-Boubaker S. SARS-CoV-2 infection in pediatric population before and during the Delta (B.1.617.2) and Omicron (B.1.1.529) variants era. Virol J. 2022;19(1):144.) but have expanded added the new suggested reference.

2) In the Methods-Data and Sample Collection, did the authors collect data on socioeconomic status of the 258 household “clusters”? Higher socioeconomic status is generally associated with lower SARS-CoV-2 antibody prevalence [Naeimi et al. eClinical Medicine 2023;56:101786].

We did not. This has also been suggested by other reviewers. We clarified this point in the methodology section.

3) Similarly, do we know how many individuals were diabetic, since the condition is associated with higher ACE2 expression in many tissues (lungs, kidneys, pancreas, etc)? Conversely, patients with allergic sensitization and asthma have lower ACE2 gene expression [Shukla. Eur Arch Otorhinolaryngol 2020;278:2637-40].

We thank the reviewer for highlighting the potential impact of specific comorbidities such as diabetes and allergic conditions on ACE2 expression. We did collect data on the presence of chronic comorbidities, including diabetes and asthma. However, due to the limited number of individuals with each specific condition, we grouped comorbidities into

---

## [Decision Letter · Decision Letter 1]

Age-dependent ACE2/TMPRSS2 expression and SARS-CoV-2 household transmission in Gran Canaria

PONE-D-25-17289R1

Dear Dr. nóvoa-medina,

We’re pleased to inform you that your manuscript has been judged scientifically suitable for publication and will be formally accepted for publication once it meets all outstanding technical requirements.

Kind regards,

Mohamed Samy Abousenna, Ph.D

Academic Editor

PLOS ONE

Additional Editor Comments (optional):

Reviewers' comments:

Reviewer's Responses to Questions

**Comments to the Author**

Reviewer #1: All comments have been addressed

Reviewer #4: All comments have been addressed

Reviewer #5: All comments have been addressed

2. Is the manuscript technically sound, and do the data support the conclusions?

Reviewer #1: Yes

Reviewer #4: Yes

Reviewer #5: Yes

3. Has the statistical analysis been performed appropriately and rigorously?

Reviewer #1: Yes

Reviewer #4: Yes

Reviewer #5: (No Response)

4. Have the authors made all data underlying the findings in their manuscript fully available?

Reviewer #1: Yes

Reviewer #4: No

Reviewer #5: No

5. Is the manuscript presented in an intelligible fashion and written in standard English?

Reviewer #1: Yes

Reviewer #4: Yes

Reviewer #5: Yes

Reviewer #1: The authors responded and made all necessary changes. So this manuscript can nor accepted as a publication.

Reviewer #4: (No Response)

Reviewer #5: The authors have addressed my comments and have overall improved the paper. However, before publishing, all graph axes should be labeled, graphs titled, and legends for every single graph.

**Do you want your identity to be public for this peer review?** For information about this choice, including consent withdrawal, please see our Privacy Policy

Reviewer #1: **Yes: ** Mahmut Cerkez Ergoren

Reviewer #4: **Yes: ** Fernando Namuche

Reviewer #5: No

---

## [Editor Report · Acceptance letter]

PONE-D-25-17289R1

PLOS ONE

Dear Dr. nóvoa-medina,

I'm pleased to inform you that your manuscript has been deemed suitable for publication in PLOS ONE. Congratulations! Your manuscript is now being handed over to our production team.

Kind regards,

on behalf of

Dr. Mohamed Samy Abousenna

Academic Editor

PLOS ONE